# Scots Pines (*Pinus sylvestris*) as Sources of Biological Ice-Nucleating Macromolecules (INMs)



**Teresa M. Seifried** [1,2,†], **Florian Reyzek** [1,†], **Paul Bieber** [1,2] **and Hinrich Grothe** [1,*]

1    Institute of Materials Chemistry, TU Wien, Vienna 1060, Austria
2    Chemistry Department, University of British Columbia, Vancouver, BC V6T 1Z1, Canada
*    Correspondence: hinrich.grothe@tuwien.ac.at
†    These authors contributed equally to this work.

**Abstract:** Scots pine (*Pinus sylvestris*) is the most widespread pine species in the world. It grows in the largest forest system in the northern hemisphere and, together with birch trees, occupies a major part of the boreal forests. Recently, birch trees have been discovered as important emission sources of terrestrial ice-nucleating macromolecules (INMs) coming from pollen, bark, leaves, petioles, branches, and stem surfaces. It is known that pine pollen nucleate ice; however, the potential of other tree parts releasing INMs and contributing to the emission of ice-active aerosols is unknown. Here, we investigated the distribution of INMs in, on, and around Scots pines (*Pinus sylvestris*) in a laboratory and field study. We collected bark, branch wood, and needle samples from six pine trees in an urban park in Vienna, Austria. The concentration of INMs from aqueous extracts of milled (powder extracts) and intact surfaces (surface extracts) were determined. In addition, we collected rainwater rinsed off from three pines during a rainfall event and analyzed its INM content. All investigated samples contained INMs with freezing onset temperatures ranging from $-16\ °C$ to $-29\ °C$. The number concentration of INMs in powder extracts at $-25\ °C$ ($n_{INMs}(-25\ °C)$) ranged from $10^5$ to $10^9$ per mg dry weight. Surface extracts showed concentrations from $10^5$ to $10^8$ INMs per $cm^2$ of extracted surface, with needle samples exhibiting the lowest concentrations. In the rain samples, we found $10^6$ and $10^7$ INMs per $cm^2$ of rain-collector area at $-25\ °C$, with freezing onset temperatures similar to those observed in powder and surface extracts. With our data, we estimate that one square meter of pine stand can release about $4.1 \times 10^9$ to $4.6 \times 10^{12}$ INMs active at $-25\ °C$ and higher, revealing pine forests as an extensive reservoir of INMs. Since pines are evergreen and release INMs not only from pollen grains, pines and the boreal forest in general need to be considered as a dominant source of INMs in high latitude and high-altitude locations, where other species are rare and other ice nuclei transported over long distances are diluted. Finally, we propose pine trees as an INM emission source which can trigger immersion freezing events in cloud droplets at moderate supercooled temperatures and therefore may have a significant impact on altering mixed phase clouds.

**Keywords:** biological ice nucleation; pine trees; heterogeneous ice nucleation; INMs; bioaerosols





## 1. Introduction

Scots pine (*Pinus sylvestris*) is one of the most widely distributed pine species in the world [1]. It predominantly grows in large stands in the boreal forest, which covers about 11% of the overall land surface on Earth [1]. The boreal vegetation is among the strongest emitters of bioaerosols—a diverse and complex classification of aerosols consisting of viable and non-viable classes of biological material [2], e.g., viruses, bacteria, plant fragments, excretions, and pollen grains [3]. The aerodynamic diameter of bioaerosols ranges from a few nanometers up to several micrometers [3]. Various tree species release bioaerosols with essential impacts on the life cycle of many organisms and ecosystems [3]. Those bioaerosols allow genetic exchange between habitats [3]. Fertilization of seed plants (as in *Pinus sylvestris*), for example, relies on pollen grains, i.e., male gametophytes, passing

through the air to reach the female plant part during the pollination season, thus serving to reproduce the tree [4]. Bioaerosols travel long distances in the atmosphere, cross geographic boundaries and reach remote locations [5]. When bioaerosols are lifted to high altitudes in the troposphere, they may influence the evolution of ice in mixed-phase clouds by acting as ice nuclei (IN) [6]. In general, ice formation in cloud droplets occurs either homogenously at temperatures around $-38$ °C or heterogeneously with the presence of IN at warmer temperatures [7]. Cloud glaciation affects the microphysical state of a cloud and thus its radiative properties, precipitation patterns, and cloud lifetime [8]. The presence of ice in the atmosphere strongly influences the radiative balance of planet Earth [9–11]. More than 50% of global precipitation originates from the ice phase [12]. However, the role of ice in clouds and the resulting implications on the climate system remains uncertain. An advance in atmospheric ice nucleation research is needed to improve the understanding of the influence of cloud glaciation on the radiation balance. Past studies have pointed to the fact that mineral dust is the dominant IN in the regime of mixed-phase clouds [13–15]. Nevertheless, the presence of inorganic IN alone cannot explain glaciation of clouds at warmer temperatures and it is assumed that bioaerosols contribute to the freezing process [16]. In fact, many studies in boreal and arctic regions observed high biological IN concentrations in the lower troposphere. Wex et al. (2019) [17] showed that biological IN concentrations in different arctic locations peak from late spring to fall and suggested open land and water as sources for these aerosol particles.

Boreal [18] and alpine forests [19] are considered to contribute to the emission of biological IN. Challenging growing conditions such as cold temperatures, short growing seasons, and permafrost, resulting in simple woody vegetation dominated by a few cold-hardy species [1] characterize those forests. Among them are Scots pines which use extracellular freezing [20–22] to survive sub-zero temperatures. Extracellular freezing in frost-tolerant woody plants occurs in various tissues [20,23].Ice-nucleating substances trigger ice crystal formation in extracellular spaces that allow the withdrawal of water from the cell [23]. This process leads to an increase in the supercooling capacity of the cell, preventing it from being damaged by ice crystals [24]. Thus, we hypothesize that Scots pines inherited IN for survival reasons, and we assume that this is a property found throughout this species no matter the growing region as shown for birch trees by Felgitsch et al., 2018 [25]. They found no significant difference between the IN concentrations of samples from alpine and urban areas. In 1966, Soulage [26] was the first to suggest pine forest ecosystems act as a local emission source of atmospheric IN. In lab experiments, we could show that pine pollen (*Pinus sylvestris*) act as IN [27]. The exceptional feature of the ice nucleation activity of pollen lies in the fact that not the whole grain but suspendable macromolecules (ice-nucleating macromolecules, INMs) that are easily extractable from the grain's surface with water induce heterogeneous ice nucleation [28]. Recently, it was found by us that in case of birch pollen grains, which are also ice nucleation active, not only the pollen but also bark, wood, and leaves contain INMs [25,29] and are released into the environment during rainfall [19]. The small size of INMs increases their mobility allowing them to be transported far distances. A recent study by Paramonov et al., 2020 [18] in Southern Finland correlated an increase in ice-nucleating particle concentration over the boreal environment with sub-0.1 µm biological fragments such as INMs. In addition, INMs from birch trees, which are growing in boreal forests as well, are released to the surrounding environment evoked by rain events [19], getting airborne and transported vertically. Thus, INMs from boreal vegetation could in fact contribute to microphysical processes in the lower troposphere. While the formation and transport of INMs from birches have already been studied [19,25,27,29,30], respective studies on the same level for pines are still missing. This knowledge gap limits regional and global estimations of the boreal forest acting as an emission source of INMs. Furthermore, the inclusion of INMs from boreal forests into climate calculations and simulations is hindered without data on the concentration and release of INMs from Scots pines.

In this study, we present laboratory and field experiments, where we investigated the ability of Scots pines to serve as a source of INMs. We first quantified the distribution of INMs among six different Scots pines in the laboratory by extracting INMs from milled powder and intact tissue surfaces and measuring ice nucleation in immersion freezing mode. We aimed to quantify the general INM content of this tree species and assume that it is independent of growing location, as previously described by Felgitsch et al., 2018 [25] for birch trees. Thereafter, we investigated whether INMs enter the surrounding environment of the trees during a representative rainfall event by sampling rainwater wash-off in the field. The objectives of this study were (i) to investigate the nucleation temperatures and concentrations of INMs in powder extracts of bark, branch wood, and needle tissues, (ii) quantify the INM content on the surfaces, respectively, and compare the concentration to bulk extracts to investigate whether INMs originate from the tree and (iii) investigate if rain washes down INMs from the trees, releasing ice-active material to the environment.

## 2. Materials and Methods

### 2.1. Sample Acquisition

We collected tissues from six mature Scots pines (*Pinus sylvestris*), named alphabetically (Pine A to F) in different parks in Vienna, Austria (Figure 1). Pine A is located in a park close to the highway. Pine B and C grow in the middle of the largest park in Vienna (Prater). Pine D and E grow next to the railroad line and F at a school yard in the 22nd district of Vienna. Table 1 summarizes the detailed information on all trees including sampling time, location, altitude, and stem perimeters. Figure A1 in the appendix shows images of all trees that were investigated in this study. From each tree, we collected bark, main branch wood, and needle samples. All tools used during sampling were disinfected and cleaned with ethanol (approx. 90 vol. %) before and after each usage. Samples were frozen at −20 °C within a few hours of sampling and stored at that temperature until further sample preparation.

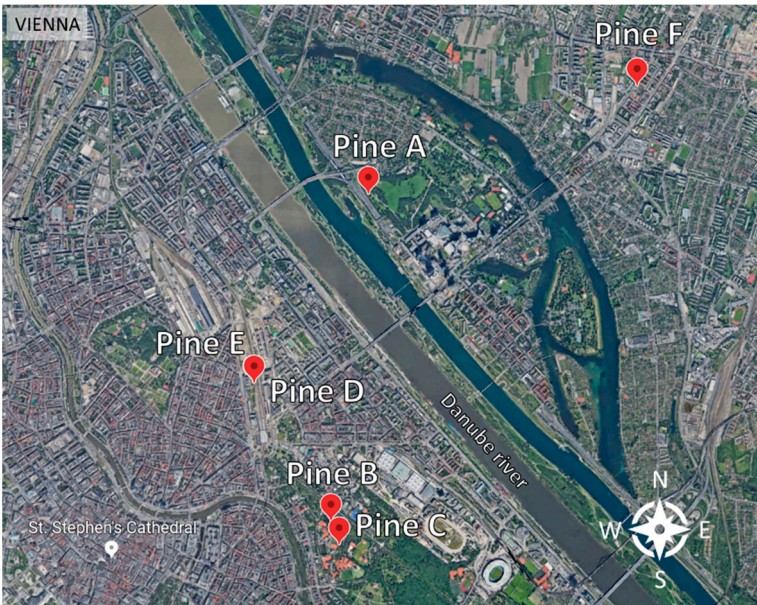

**Figure 1.** Map of Vienna showing all sampling locations (Pine A to Pine F) with red tags. Pine E and D grow directly next to each other. Image adapted from Google Earth ©, https://earth.google.com/, (accessed on 28 June 2022).

**Table 1.** Information of the sampling date, location, altitude, and circumferences at 1 m of the corresponding tree. * The trees are listed in the tree cadastre of the city of Vienna (https://www.wien.gv.at/umweltgut/public/, accessed on 27 January 2020). The tree number is ambiguous but combined with the species, trees can be identified.

| Sample ID | Collection Date | GPS Waypoints Longitude, Latitude (°) | Altitude (m) | Circumference of Trunk at 1 m (cm) | Tree Cadastre * | Weather Conditions |
|---|---|---|---|---|---|---|
| Pine A | 28 January 2020 | 48.238220, 16.405210 | 166 | 52 | 13088A | rainy |
| Pine B | 14 February 2020 | 48.211290, 16.400590 | 163 | 68 | 226 | sunny, windy |
| Pine C | 14 February 2020 | 48.209360, 16.401601 | 161 | 122 | 187 | sunny, windy |
| Pine D | 14 February 2020 | 48.222680, 16.391030 | 169 | 62 | 12 | sunny, windy |
| Pine E | 14 February 2020 | 48.222680, 16.391090 | 169 | 60 | 11 | sunny, windy |
| Pine F | 15 February 2020 | 48.247150, 16.438460 | 163 | 78 | 28 | cloudy, drizzle |

*2.2. Sample Preparation*

2.2.1. Powder Extracts

We obtained powder extracts in accordance with Felgitsch et al. (2018) [25] and Seifried et al. (2020) [29]. In short, tree tissue samples were first cryo-milled with a swing mill (MM400, RETSCH, Haan, Germany). The resulting powder was then dried, extracted with ultra-pure water and filtered prior to ice nucleation analysis. In more detail, a sample (e.g., ~20 needles) was placed together with a steel ball in a steel sample holder. The container was tightly sealed and inserted into a liquid nitrogen bath to cool the sample and prevent frictional heat from altering any substances and biomolecules during the grinding process. Cooling lasted for 2 min prior to the milling procedure and 1 min between each grinding step. Each sample was milled four times with a grinding interval lasting 30 s at a frequency of 25 Hz. The powder obtained was then dried over silica gel in a desiccator until the sample reached constant weight. Afterwards, 50 mg of dry powder were weighed into an Eppendorf tube and carefully mixed with 1 mL ultra-pure water. The suspension was allowed to stand for about 6 h and shaken three to four times. This was followed by centrifugation (2-16P, Sigma, Albuch, Germany) for 5 min at 1123 rcf (relative centrifugal force), and the extract was filtered with a sterile 0.2 µm syringe filter (cellulose acetate membrane, sterile, VWR International, Radnor, PA, USA). Prior to ice nucleation measurements, samples were stored at −20 °C. Figure A2 in the Appendix A gives an overview of the milling procedure and preparation of powder extract samples.

2.2.2. Surface Extracts

INMs from the surface of intact pine tissues were extracted in ultra-pure water for 6 h. Each sample was centrifuged and filtered with a 0.2 µm syringe filter (cellulose acetate membrane, sterile, VWR International, USA) and stored in an Eppendorf tube until the freezing experiments. In more detail, branch wood pieces were cut in 1 to 10 cm sections. The cut edges were covered with paraffin wax (Sigma-Aldrich, St. Louis, MO, USA) to prevent sap leakage. Note that the sealing wax was measured for ice nucleation activity and found to be inactive. The branch wood samples were then immersed in ultra-pure water (water volume varied between 1 mL and 9 mL depending on the sample size). Needles were only half immersed in ultra-pure water, to avoid extraction of the branch wood, from which the needles are growing out of. Bark samples were embedded in wax in a petri-dish and then extracted with ultra-pure water. Figure 2 depicts the samples during the extraction process.

To calculate the extracted surface area, we estimated all samples as geometric figures. Note that for simplicity reasons, surface roughness was not considered when estimating the area of each sample surface. Branch wood samples were modelled as cylinders (see Figure A3) and for bark samples, the not wax-covered part of the surface was estimated as planar rectangle (see Figure A4). To calculate the surface of a single pine needle, we first measured the length that was in contact with water. Secondly, we recorded microscopic images of needles embedded in wax and measured the perimeter using ImageJ

(see Figure A5). The height of the immersed part of the needle and the average perimeter served as variables to calculate the surface of each needle. Table 2 gives the surfaces per extraction volume used to calculate $n_{INMs}(T)$ according to Equation (3).

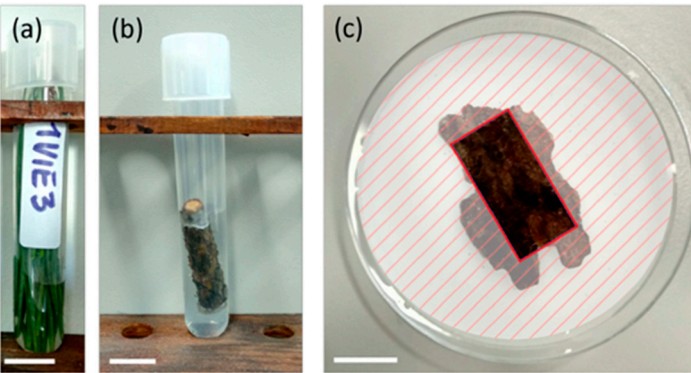

**Figure 2.** Images of (**a**) needles, (**b**) branch wood, and (**c**) bark from Scots pines immersed in ultra-pure water to extract INMs from the surface. The shaded area in (**c**) indicates the parts of bark, which were embedded in paraffin wax. The red rectangle represents the area which was used to calculate the surface of extraction. The white bar in (**a**–**c**) corresponds to 10 mm.

**Table 2.** Surface area per extraction volume (ultra-pure water) of pine needles, branch wood, and bark samples used to calculate the cumulative ice nuclei concentration $n_{INMs}(T)$.

| Sample ID | Needles ($cm^2mL^{-1}$) | Branch ($cm^2mL^{-1}$) | Bark ($cm^2mL^{-1}$) |
|---|---|---|---|
| Pine A | 0.43 | 1.22 | 10.05 |
| Pine B | 1.43 | 1.14 | 15.65 |
| Pine C | 0.58 | 0.70 | 10.62 |
| Pine D | 1.08 | 0.52 | 14.03 |
| Pine E | 3.21 | 0.34 | 8.11 |
| Pine F | 1.25 | 0.77 | 9.55 |

### 2.2.3. Rain Samples

We collected rain underneath three pines (Pine A, B, and C) overnight from 1 March to 2 March 2020 using self-built rain-collectors. Detailed information on the collectors is given in Seifried et al. (2020) [29]. Briefly, a rain-collector consists of a sterile centrifuge tube (polypropylene, 50 mL, Brand, Germany) mounted on a wooden pole, which is anchored to the ground with three guide ropes. We placed the collectors underneath each pine. The rain collectors were placed at 1 to 3 m from the trunk of the trees so that the rain droplets interacted with the tree crown before landing in the collector. Pure rainwater, which was collected 5 m away from the trees served as a blank control (see Figure A6). After the rainfall event, we recorded the total volume of each sample and stored the samples at −20 °C. In the laboratory, samples were thawed and an aliquot of 1 mL of each sample was filtered with a 0.2 μm syringe filter (cellulose acetate membrane, sterile, VWR International, USA) prior ice nucleation activity measurements.

### 2.3. Ice Nucleation Assay

All freezing experiments were performed in immersion freezing mode using the cryo-microscopy setup VODCA (Vienna Optical Droplet Crystallization Analyzer). Felgitsch et al. (2018) [25] provides a detailed description of the setup. Briefly, the freezing assay consists of two main components: an incident light microscope (BX51M, Olympus, Tokyo, Japan) with an attached camera (MDC320, Hengtech, Germany) linked to a computer and a cryo-cell. The cryo-cell is a polymer-based compartment that can be closed airtight. It contains a cooling unit consisting of a Peltier element (Quick-Cool QC-31-1.4-3.7M) with a

thermocouple (IEC K, R&S) fixed on top and a heat exchanger, cooling the warm side of the Peltier element during freezing experiments. Before conducting a freezing experiment, the samples are incubated at room temperature for approximately 15 min. The samples are all measured as aqueous components of an emulsion created on a clean glass slide, which is placed on top of the Peltier element. A LabVIEW-based software enables to record videos during the freezing process. All freezing experiments were performed with a cooling rate of $10\ °\mathrm{C\,min}^{-1}$. Only droplets in the size range between 15 and 40 µm (droplet volume: 1.8–34 pL) were included in our evaluations. Ultra-pure water of MilliQ grade (18.2 MΩ·cm, Millipore® SAS SIMSV001, Merck Millipore, Burlington, MA, USA) served as a reference for homogeneous freezing events and froze below $-34\ °\mathrm{C}$. Highly ice nucleation active samples were diluted with ultra-pure water to avoid underestimations of INM concentrations. All dilutions were prepared by adding an appropriate volume of sample stock solutions to ultra-pure water to reach a total of 1000 µL (e.g., 10 µL of sample stock solution to 990 µL of ultra-pure water for a 1:100 dilution).

Data Analysis

The number of ice-active substances above a certain temperature can be expressed by the cumulative nucleus concentration $n_{INMs}(T)$, assuming ice nucleation to be a time-independent process [31,32], and calculates as follows:

$$n_{INMs}(T) = -\frac{\ln(1 - f_{ice}(T))}{V_{droplet}} \cdot D \tag{1}$$

where $f_{ice}(T)$ represents the fraction of frozen droplets, which is the number of droplets frozen at a certain temperature divided by the total number of droplets analyzed in the experiment. $V_{droplet}$ accounts for the average droplet volume of the freezing assay (8.2 pL using VODCA), and $D$ is the dilution factor of analyzed solutions.

To refer the number concentration of INMs per volume to the sample's surfaces, we modified $n_{INMs}(T)$ by multiplying with the extraction volume, $V_{extraction}$ divided by the surface of the sample, $\sigma_{sample}$ [29]:

$$n_{INMs}(T) = -\frac{\ln(1 - f_{ice}) \cdot D}{V_{droplet}} \cdot \frac{V_{extraction}}{\sigma_{sample}} \tag{2}$$

To estimate the area of the sample surfaces, we used approximations as indicated in Section 2.2.

Equation (3) was used to calculate INM concentrations extracted from the pines during rainfall events [29]. $n_{INMs}(T)$ from Equation (1) was modified by multiplying with the rain volume, $V_{rain}$ divided by the area of the precipitation collector's inlet, $\sigma_{inlet}$ (circular):

$$n_{INMs}(T) = -\frac{\ln(1 - f_{ice}) \cdot D}{V_{droplet}} \cdot \frac{V_{rain}}{\sigma_{inlet}} \tag{3}$$

This modification allowed for referring the concentration of INMs to pines that are exposed to rainfall.

Note that although many assumptions were included in the calculations, the cumulative number of INMs normally spans over an exponential range and small deviations in surface area or droplet volumes do not greatly influence the final number concentrations of INMs.

## 3. Results

### 3.1. INM Distribution in Bark, Branch Wood and Needle Powder Extracts

In total, all 18 powder extracts from bark, branch wood, and needles showed ice nucleation activity with $n_{INMs}(T)$ values ranging from $2.4 \times 10^5$ (Pine F, bark) to $1.8 \times 10^9$ (Pine C, needles) per mg dry weight. Figure 3 compares the cumulative spectra of all trees

per sample category from a dilution where about 50% of droplets froze heterogeneously[25]. We chose a specific dilution for every sample based on its ice nucleation activity to avoid underestimations of INMs number concentrations [25]: a too highly concentrated sample would contain too many INMs per droplet and therefore the calculated concentrations would underrepresent the true concentrations. The dilutions ranged from 1:10 to 1:1000 ($v/v$) from the least active to the most active sample. Most samples showed a significant increase in INMs number concentration in the temperature range from $-20\,^\circ$C to $-25\,^\circ$C. Therefore, we chose to compare $n_{INMs}(-25\,^\circ$C) values. In general, branch wood and needle samples exhibited higher concentrations compared to bark. The $n_{INMs}(-25\,^\circ$C) values of needles ranged from $9.8\times10^6$ mg$^{-1}$ (Pine F) to $1.8\times10^9$ mg$^{-1}$ (Pine C). The branch wood samples showed similar values but also scattered to lower values. Bark extracts showed the lowest amount of INMs per mg. The curves scatter strongly, and the $n_{INMs}(-25\,^\circ$C) values ranged between $2.4\times10^5$ mg$^{-1}$ (Pine F) and $4.9\times10^8$ mg$^{-1}$(Pine E). The concentration of INMs in needles is lower when comparing younger trees (Pine D, E and F) with older trees (Pine A, B and C). The reason for this observation could be that older needles are enriched with INMs compared to freshly grown needles. Furthermore, the onset temperature ($T_{on}$) of the undiluted samples ranged from $-17.2\,^\circ$C (Pine C, branch wood) to $-28.6\,^\circ$C (Pine D, bark), not including the branch wood extract of Pine B, which started freezing at $-13.2\,^\circ$C (see Figure A7). The bark samples generally showed the lowest $T_{on}$, while branch wood and needle samples exhibited similar $T_{on}$. All $n_{INMs}(-25\,^\circ$C) and $n_{INMs}(-34\,^\circ$C) values of the shown data in Figure 3 are summarized in Table A1 in the Appendix B.

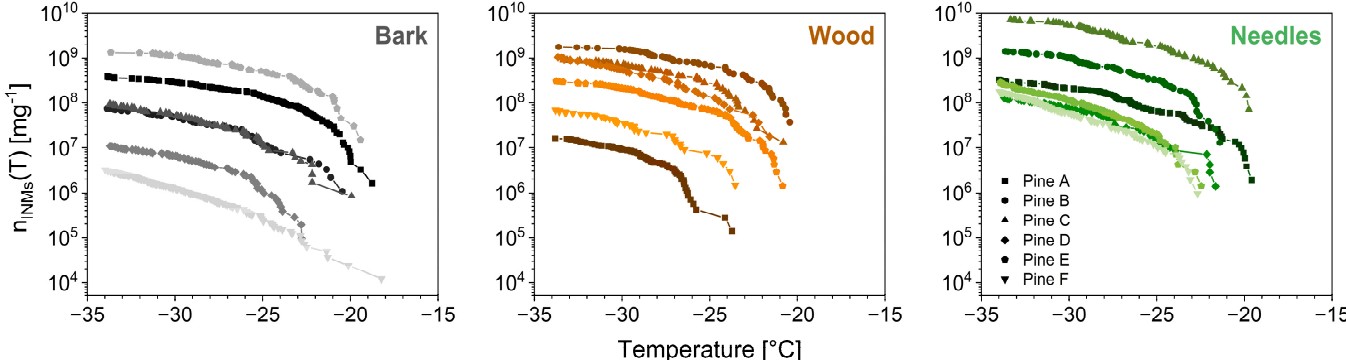

**Figure 3.** Cumulative INM concentrations, $n_{INMs}(T)$ per extracted powder mass of bark, branch wood and needle samples from six Scots pines (*Pinus sylvestris*). Following dilutions are depicted in the graphs: Bark—Pine F undiluted; Pine D 1:10; Pine A, B, C 1:100; Pine E 1:1000. Branch wood—Pine A 1:10; Pine E, F 1:100; Pine B, C, D 1:1000. Needles—Pine A, D, E, F 1:100; Pine B 1:1000; Pine C 1:5000. All $n_{INMs}$ at $-25\,^\circ$C and $-34\,^\circ$C of the shown data are summarized in Table A1 in the Appendix B.

### 3.2. INM Distribution on the Surface of Bark, Branch Wood and Needles

Surface extracts represent the amount of INMs that can be washed down with ultra-pure water from the surface of a corresponding intact tree tissue. All surface extracts froze heterogeneously with $T_{on}$ rather similar to powder extracts (see Figures 4 and A8), varying between $-17.4\,^\circ$C (Pine A, branch wood) and $-25.1\,^\circ$C (Pine D, bark). $n_{INMs}(-25\,^\circ$C) values ranged from $5.8\times10^4$ (Pine B, needles) to $2.8\times10^8$ (Pine A, branch wood) per cm$^2$. Bark and branch wood surfaces generally provided higher INM concentrations per surface area than needles. Needle extracts exhibited the lowest $n_{INMs}(-25\,^\circ$C) values per cm$^2$. This trend is dominant in all pines. In addition, only a few droplets from Pine D and E needles froze in the heterogeneous region. An influence of the weather conditions (see Table 1) on the IN concentration was not observed.

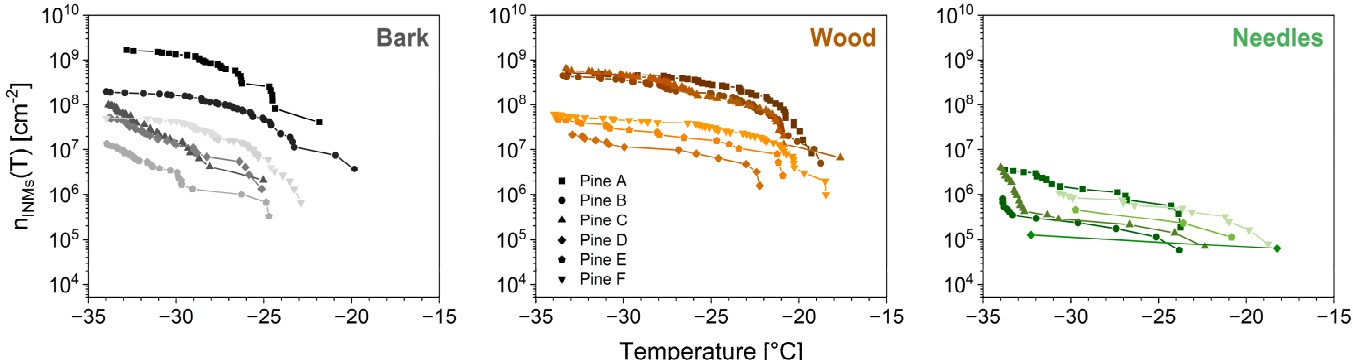

**Figure 4.** Cumulative INM concentrations, $n_{INMs}(T)$ per extracted surface of bark, branch wood, and needle extracts from six Scots pines (*Pinus sylvestris*). Following dilutions are depicted in the graphs: Bark—Pine C, D, E, F undiluted; Pine B 1:5; Pine A 1:10. Branch wood—Pine D, E, F undiluted; Pine B, C 1:5; Pine A 1:10. Needles—Pine A to F undiluted. All $n_{INMs}$ at $-25\,°C$ and $-34\,°C$ of the shown data are summarized in Table A2 in the Appendix B.

### 3.3. The Effect of Precipitation on the Release of INMs from Pines

Rain samples collected underneath the trees' canopy were assessed for ice nucleation activity. All rain samples showed heterogeneous freezing, i.e., all droplets froze above the reference water blank ($>-34\,°C$; see Figure A9). Like powder and surface samples, $T_{on}$ ranged between $-16.2\,°C$ and $-23.8\,°C$. The course of cumulative freezing spectra of all rain samples are rather similar to each other (Figure 5). Comparing the spectra with the laboratory extracts (Figures 3 and 4) we recognized that the increase in INMs concentration between $-20\,°C$ and $-25\,°C$ was less pronounced for the rainfall extractions. The number of active INMs above $-25\,°C$ were in the order of magnitudes between $10^6$ and $10^7\,cm^{-2}$ (giving the area of the rain collector inlet). The lowest concentration was measured for Pine B, sampler #2 with $5.3 \times 10^5\,cm^{-2}$. Pine C sampler #2 showed the highest concentration with $1.7 \times 10^7\,cm^{-2}$. In addition, blank samples which were set up next to the trees froze homogeneously, except for one droplet at $-30\,°C$ (see Figure A9).

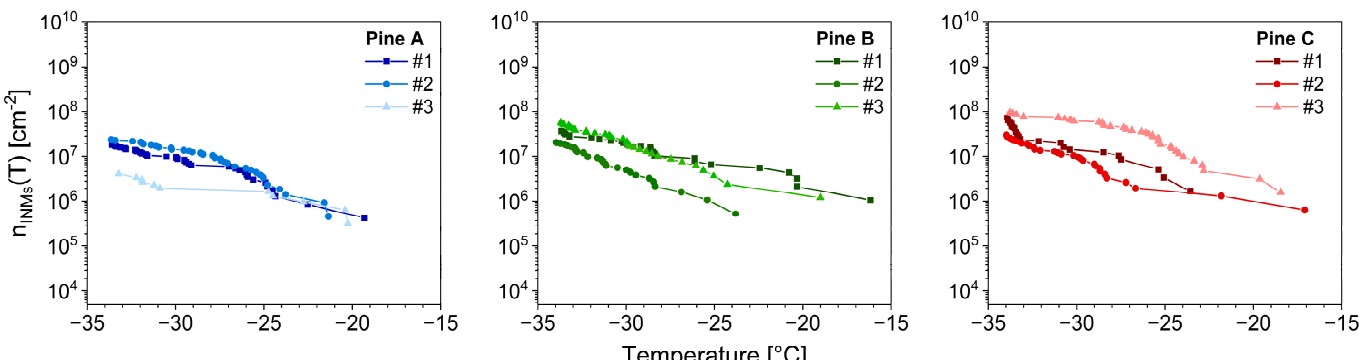

**Figure 5.** Cumulative INM spectra, $n_{INMs}(T)$ of collected rain samples (three samples per tree, named #1, #2 and #3) underneath Pine A, B, and C. All $n_{INMs}$ at $-25\,°C$ and $-34\,°C$ of the shown data are listed in Table A3 in the Appendix B.

## 4. Discussion

This study elucidates Scots pines (*Pinus sylvestris*) as a terrestrial reservoir and potential emission source of INMs. We found ice nucleation active material (<200 nm) on all investigated pine surfaces. Our results show that precipitation can wash INMs off the tree's surface. The latter is essential to better understand the release process of INMs into the atmosphere.

In 2002, Diehl et al. (2001) [33] showed that pollen of Scots pine (*Pinus sylvestris*) nucleate ice. However, their atmospheric relevance remains questionable due to the large

size of pollen grains, and hence a short residence time in the atmosphere [34,35]. However, Pummer et al. (2012) [27] extracted and filtered macromolecules from pollen and proved that the ice nucleation activity remains the same when the pollen grains were removed from the suspension. Due to their small size (<200 nm), INMs from pollen can be easily lifted to high altitudes, possibly using sub-pollen particles, pollen fragments, or droplets as their transport vehicles [36,37]. Furthermore, studies found INMs all over the tissues of different birch trees (*Betula pendula*) [25,29], providing proof that INMs are much more abundant, both special and seasonal, than previously thought. However, similar studies on different tree species were missing, and therefore, it is impossible to evaluate alpine and boreal mixed forests as global emission sources of INMs.

The inhere investigated Scots pines grow in Vienna. We assume that these pines represent Scots pines in general, since Felgitsch et al., 2018 [25] found no significant difference between INM concentrations of tree samples from alpine and urban areas. INMs were each extracted from the bulk (powder) and the tissues' surfaces. All samples showed ice nucleation activity with $T_{on}$ between $-17\,°C$ and $-28\,°C$. The previously reported $T_{on}$ of pine pollen between $-17\,°C$ and $-20\,°C$ [27,30,33] is within this range. Interestingly, a single sample, namely branch wood from Pine B, showed a $T_{on}$ of $-13.2\,°C$ (see Figure A7), suggesting that in INMs, mixtures of different biological compounds (polysaccharides, proteins, etc.) might exist, some of which have $T_{on}$ higher than $-15\,°C$. For pollen grains of birch, Dreischmeier, et al. 2017 [30] have shown that the $T_{on}$ can be as high as $-8\,°C$. Since our VODCA set-up is only suitable for high INM concentrations (>$10^5$ INM per mg pollen), we conclude that low concentrations of highly active INM might exist in our pine extracts, which we have not detected. In a control experiment (see Figure A10), we could show that the $T_{on}$ of pine pollen can be as high as $-7.7\,°C$ at a concentration of $10^{-1}$ INM per mg pollen using the more sensitive TINA set-up [38]. Additionally, there might also be other INMs involved from organisms surrounding the tree's surface, e.g., from bacteria or fungal spores [28,31,39–41].

For powder extracts, we found the highest INM concentration in needles (see Figure 3), while bark and branch wood concentrations were typically an order of magnitude lower. The overall concentrations range over four orders of magnitude and within a sample type (bark, branch wood, needles), typically around three orders of magnitude. This variation can be due to age, size, and numerous other factors affecting a single tree, e.g., nutrient availability. However, no clear trend suggests a single tree having higher INM concentrations over the others.

In contrast, the number of INMs extracted from the surface is the lowest for needles. Possibly due to the hydrophobic wax covering of pine needles [42]. However, we must mention that the surface estimation for the needles is substantially more challenging, possibly bringing a systematic error. In addition, many other factors could influence the INM concentration, such as weather prior to sampling, season, the age of the tree, cardinal direction of collected samples, or daylight access, just to name a few. A clear trend, however, was not observed.

Most interestingly, we found ice nucleation activity (INA) in rainwater collected directly underneath the trees. $T_{on}$ of the found INMs extended between $-16.2\,°C$ and $-23.8\,°C$, hence in the same range as the extracted pine INMs. No INA was found in the two blank samplers, which suggests that even the short contact between a rain droplet and the pine surface is sufficient to wash down INMs. INMs extracted by rain nucleated ice over a larger temperature range compared to laboratory extractions. We suggest that real scenario extractions lead to a more random distribution of different INMs with different nucleation temperatures. For example, a rain droplet that extracts biological material from the tree's surfaces can be in contact with needles, bark, and branches before being captured with the sample collector. This process results in suspensions with diverse INMs, resulting in less steep freezing spectra (compare Figures 4 and 5). Furthermore, rain droplets might burst upon contact with the pine surface and can release microdroplets into the environment [43], which possibly contain biological material (e.g., INMs) [44]. In addition, INMs extracted

from the vegetation can get incorporated into soil and/or land on decaying leaves, possibly contributing to the high concentrations of INMs found in leaf litter [45,46]. Only recently it has been shown that INMs found in leaf litter stay active for over 50 years [45].

Scots pine is widely spread over the northern hemisphere and is Eurasia's most common pine tree [47]. By combining measured INM concentrations of our samples with biogeological data, we were able to estimate the contribution of a pine forest as an emission source of INMs: the biogeological data included tree distributions, size, and leaf area index (LAI) estimations. The LAI of *Pinus sylvestris* varies between 1.48 and 3.57, as reported by selected studies [47–50]. It describes the one-sided leaf surface area per ground area for a particular tree or tree stand, which is doubled to account for both sides. By combining LAI and surface extract data, we were able to calculate the minimum and maximum number of extractable INMs active above $-25\,^{\circ}$C for needles: $1.7 \times 10^9$ m$^{-2}$ and $4.0 \times 10^{10}$ m$^{-2}$ pine stands. In a next step, we calculated the INM contribution from pine bark. We used tree size information (tree ages between 25 and 59 years) and tree density data (1122 to 2000 ha$^{-1}$) from two studies. The pines had diameters between 11.8 cm and 18 cm and heights between 8.7 m and 15.9 m [51,52]. We estimated the tree stem with a cylindrical shape and did not account for surface roughness. By using these numbers and combining it with the surface extracts data of bark, we estimate the number of extractable INMs from pine barks (active above $-25\,^{\circ}$C) to range between $2.4 \times 10^9$ m$^{-2}$ and $4.6 \times 10^{12}$ m$^{-2}$ pine stand. In sum, this leads to INM numbers between $4.1 \times 10^9$ m$^{-2}$ and $4.6 \times 10^{12}$ m$^{-2}$ pine stand. A 2013 study estimated the area of *Pinus sylvestris* in Finland to be around 133.000 km$^2$ [53]. Assuming that the trees are, on average, of similar size to the pine stands described above, this results in extractable INM numbers between $5.5 \times 10^{20}$ and $6.2 \times 10^{23}$ active above $-25\,^{\circ}$C for Finland alone. A detailed description of all calculation steps are summarized in Appendix C.

Arguably, these estimations are quite uncertain for several reasons. The LAI is rather inaccurate since it is difficult to measure. Moreover, it is hard to account for the size and density variations. Further, due to the lack of data, we could not include the contribution of branches. Still, this estimation shows the scope of how many extractable INMs a single tree species can host. Note that the key difference between the extractable and released INMs is that the extraction time was 6 h and the interaction of a rain droplet with the pines' surface is most likely shorter. However, when comparing $n_{INMs}(-25\,^{\circ}\text{C})$ of estimated/calculated INMs in a pine stand $-4.1 \times 10^9$ m$^{-2}$ and $4.6 \times 10^{12}$ m$^{-2}$, to $n_{INMs}(-25\,^{\circ}\text{C})$ found in the collected rain $-5.3 \times 10^9$ m$^{-2}$ to $1.1 \times 10^{11}$ m$^{-2}$, the estimation lies within a close range to the measurement data.

Since the surface of birch trees (*Betula pendula*) is also a source of INMs [29], this increases the number of potential INMs that can possibly be released from vegetation into the atmosphere. Even if just a tiny fraction of these INMs is transported to the atmosphere, pines and birches could be an essential factor for atmospheric ice nucleation and a possible explanation for a large number of biological IN found in various field campaigns [17,54]. For example, Yun et al. (2022) [54] found high numbers of biological IN in the high arctic. Their source apportionment correlates residence time over land surface >50° N with the number of IN. As the snow-coverage increased, the biological IN number decreased. Another example is the field campaign by Wex et al. (2019) [17]: They found high numbers of biological IN in the arctic but could not clearly identify the sources. Thus, the land surface possibly contributes to the high number concentrations of biological IN in the arctic. Finally, we hypothesize that boreal forests are an important source for biological INMs. Boreal forests cover large areas, especially towards the northern timberline, and they consist mainly of birch and pine trees. Biological INMs can be emitted from these forests not only during pollination seasons but throughout the year when meteorological conditions are suitable (e.g., rain extracts INMs, which are aerosolized later from soil dust). Considering the contribution of the terrestrial ecosphere to INM populations around the globe, especially in high latitudes, might be key for modeling climate and climate change. Changing temperatures on the terrestrial land surface inevitably trigger the vegetation to

change, e.g., migration of cold resistant trees to higher latitudes, change in temporal snow coverages on land surfaces or anthropogenic influence on forest population, which can directly have an impact on INM emissions and thus global climate.

## 5. Conclusions

In this study, we show that rainfall extracts INMs from the surface of Scots pines (*Pinus sylvestris*). In addition, we extracted INMs from bulk and surface samples of pine bark, branch wood and needles in a laboratory experiment.

In general, the $T_{on}$ of pine tissue samples, measured with the oil-emulsion-based VODCA system, ranged from $-17\,^{\circ}$C to $-28\,^{\circ}$C, which is similar to the literature data of pollen [27,30,33]. The concentration of INMs, $n_{INMs}(-25\,^{\circ}$C$)$, extracted from the bulk and surface of pine bark, branch wood and needles ranged from $2.4 \times 10^5$ mg$^{-1}$ to $1.8 \times 10^9$ mg$^{-1}$ bulk powder and from $5.8 \times 10^5$ cm$^{-2}$ to $2.8 \times 10^8$ cm$^{-2}$ extracted surface. Within the bulk material, the highest INM concentration was found in needles. However, the surface extractable INM concentration was the lowest for this sample type, possibly due to the natural hydrophobic wax covering of the needles.

Additionally, we conducted a real scenario experiment, where we confirm that rain extracts INMs from the tree's surface, by collecting rain directly underneath the pines.

Based on our results, we estimate that one m$^2$ of pine stand releases about $4.1 \times 10^9$ to $4.6 \times 10^{12}$ INMs active above $-25\,^{\circ}$C from the tree surface, revealing pine forests as an extensive reservoir of INMs. Rainfall extracts INMs from pine tissues, which can consequently get incorporated into the hydrological cycle. There are various pathways on how these INMs can get released into the atmosphere, where they could have an impact on cloud formation. One assumed pathway is that washed-off INMs are deposited on the soil surface and strong winds near the ground may aerosolize INMs via abrasion and further transport them to higher altitudes. Probably, water soluble INM can explain the high ice nucleation activity of organic coated dust particles[6]. Another possible transport mechanism involves an aqueous film that forms during rainfall on vegetation and soil surfaces and INMs from this film aerosolize by the mechanical impact of subsequent raindrops, similar to the bioaerosol formation mechanism proposed in the literature [29,43,55,56].

Therefore, we suggest that surface extractable INMs from Scots pines (*Pinus sylvestris*) may contribute significantly to the presence of biological INMs in the atmosphere. This could be the missing link explaining the high number of biological INMs found in various field campaigns, which likely originate from the land surface, but the actual sources could not be identified entirely [17,54]. This link could then also lead to an advanced understanding of atmospheric ice nucleation.

**Author Contributions:** T.M.S., F.R., P.B. and H.G. designed the experiments. F.R. and T.M.S. collected the samples. F.R., T.M.S. and P.B. performed the experiments. T.M.S., F.R., P.B. and H.G. discussed the experiments. T.M.S. and F.R. prepared the manuscript with contributions of P.B. and H.G., H.G. acquired the funding for this work. All authors have read and agreed to the published version of the manuscript.

**Funding:** The authors acknowledge TU Wien Bibliothek for financial support through its Open Access Funding Programme. We gratefully acknowledge support by the FFG (Austrian Research Promotion Agency) for funding as part of project no. 888109 (Lab on a Drone).

**Institutional Review Board Statement:** Not applicable.

**Informed Consent Statement:** Not applicable.

**Data Availability Statement:** Data are available upon request.

**Acknowledgments:** The authors would like to thank the Municipal Department 42—Parks and Gardens (MA 42) for allowing the sampling of selected pine trees. Further, we would like to thank Janine Fröhlich-Nowoisky and Nina-Maria Kropf from Max Planck Institute for Chemistry, Mainz, Germany, for their measurements of the pine pollen using their TINA setup.

**Conflicts of Interest:** The authors declare no conflict of interest.

## Appendix A

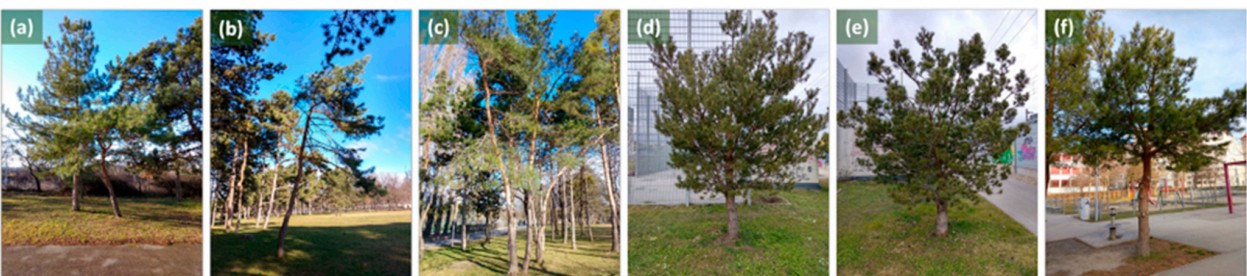

**Figure A1.** Photographs of all studied pines. (**a**–**f**) correspond to Pine A to F.

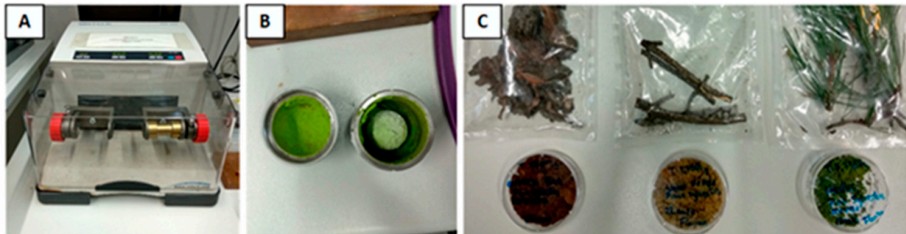

**Figure A2.** (**A**) Swing mill (MM400, Retsch, Germany) used for milling samples. (**B**) Steel container with a milled needle sample. The steel ball is visible in the bottom half of the steel container. (**C**) Wet powder from each sample type; on the left bark, in the middle branch wood, and on the right needles.

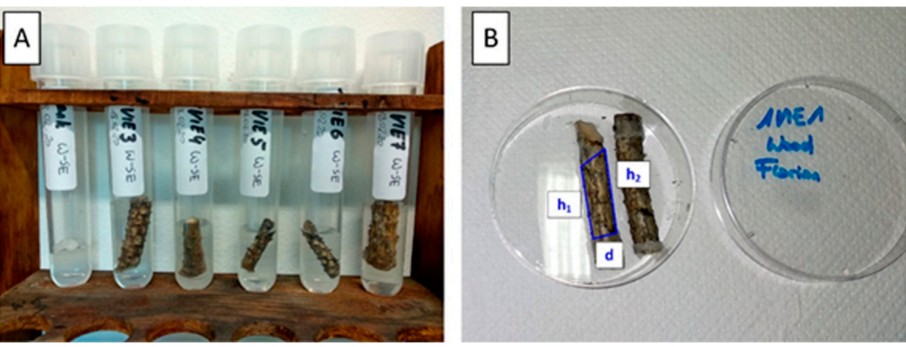

**Figure A3.** (**A**) Extraction of branch wood surface samples. The pieces of branch wood were put in a tube and filled with ultra-pure water, the tube on the left is a blank (filled with a bit of wax and ultra-pure water). Pine sample A is missing in this picture. (**B**) Branch wood sample from Pine A post extraction. The blue lines give an example of how the surface was calculated. The diameter of the branch wood as well as the longest and the shortest height of the cylinder were measured (the average height was used for the surface calculation).

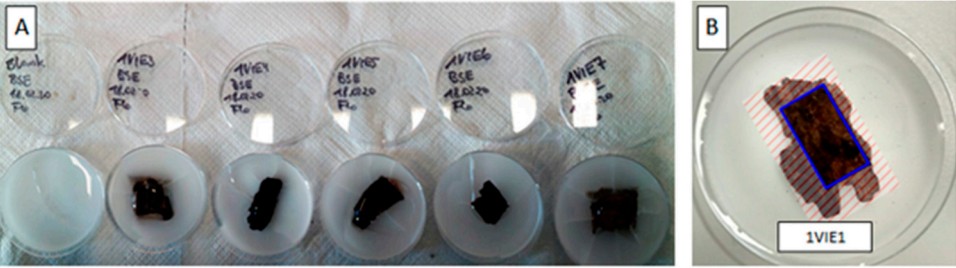

**Figure A4.** (**A**) Bark surface extracts. On the left is a blank which was prepared like a sample. (**B**) Bark sample from Pine A. The blue line marks an example of how the surface was calculated. Other parts of the bark (the hatched area) outside of the blue rectangle were not included, since there was a thin wax layer on the bark.

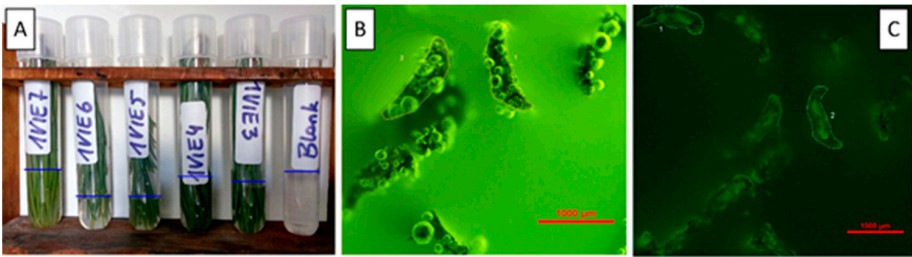

**Figure A5.** (**A**) Surface extracts of needle samples. The blue line roughly marks the water line in each tube. It is different for each sample because the length of the needles varied and we wanted to avoid that the branch wood comes into contact with the water. (**B**,**C**) Fluorescence microscopy pictures of the cross section of needles. Using the scale on the bottom right, the perimeter was measured, which can be seen by the thin white line.

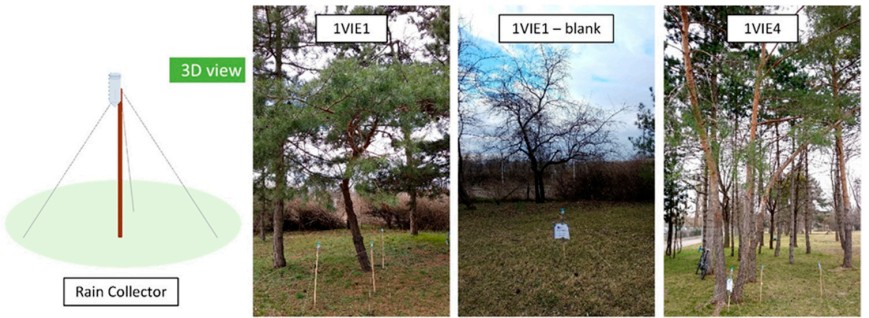

**Figure A6.** On the left is a schematic view of a rain collector. The three pictures on the right show pines with rain collectors placed underneath them. The photo in the middle shows a blank rain collector, placed in open terrain near Pine A.

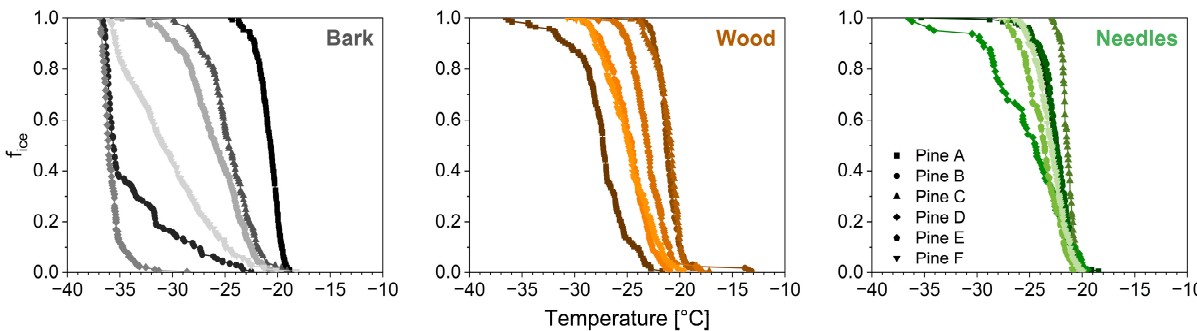

**Figure A7.** Freezing curves (fraction of frozen droplets, $f_{ice}$) of all undiluted powder extracts of bark, branch wood and needle extracts from six pines.

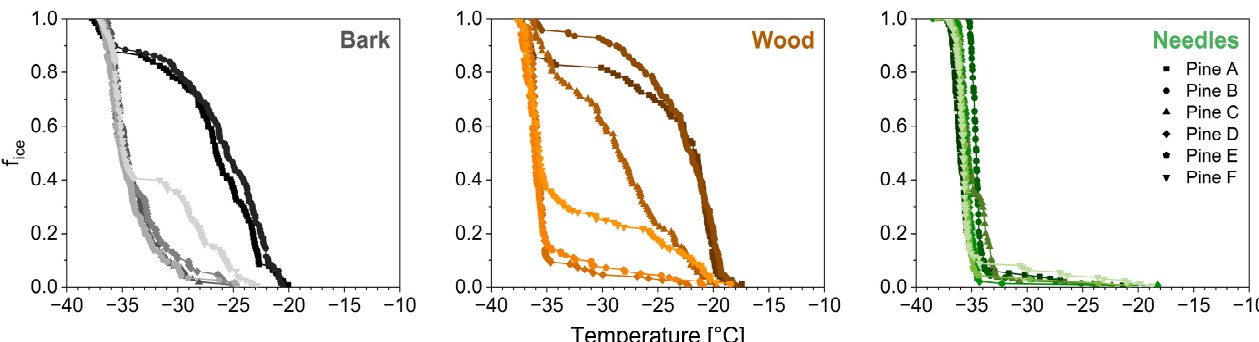

**Figure A8.** Freezing curves (fraction of frozen droplets, $f_{ice}$) of all undiluted surface extracts of bark, branch wood and needle extracts from six pines.

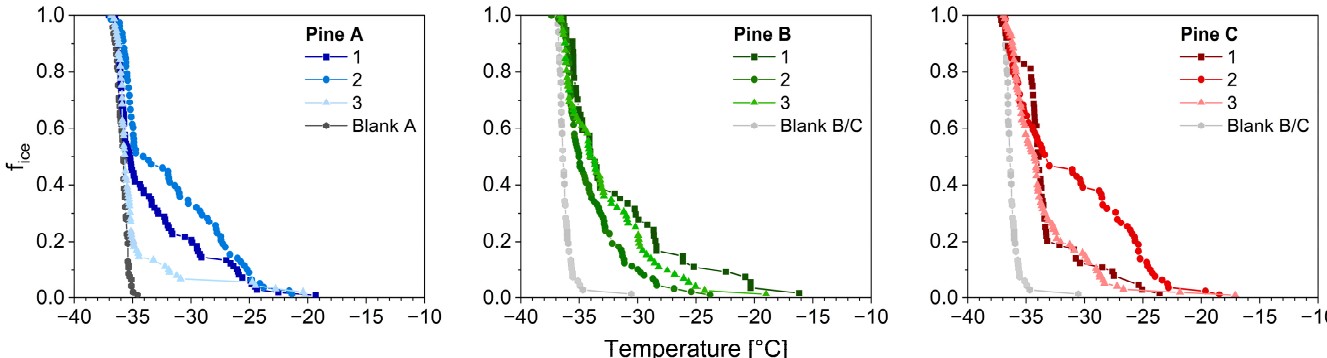

**Figure A9.** Freezing curves of the collected rain samples underneath Pine A, B, and C. Respective blank samples are plotted in grey.

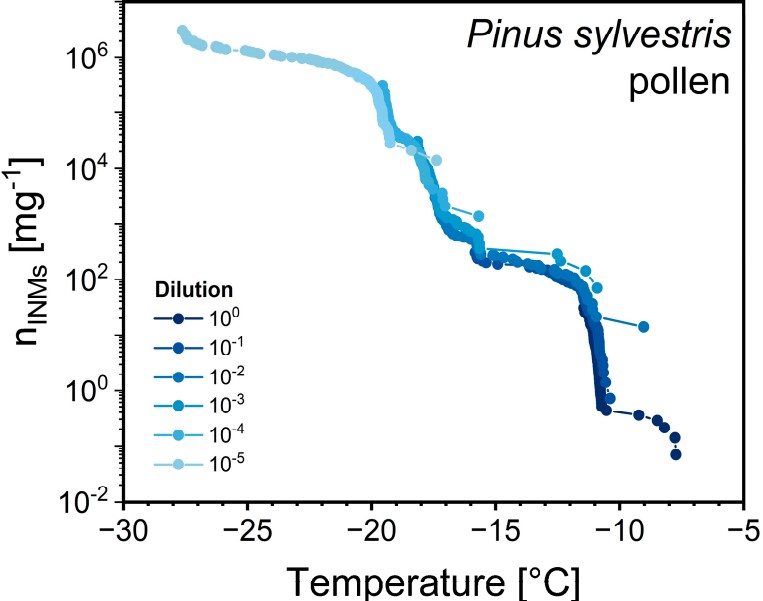

**Figure A10.** Cumulative INM spectra, $n_{INMs}(T)$ per mg pine pollen measured using TINA[38] as a control experiment.

## Appendix B

**Table A1.** INM number concentration ($n_{INMs}$) in powder extracts of bark, branch wood and needles at $-25\,^{\circ}\text{C}$ and $-34\,^{\circ}\text{C}$ (see also Figure 3).

| Pine | $n_{INMs}(-25\,^{\circ}\text{C})\,[\text{g}^{-1}]$ | | | $n_{INMs}(-34\,^{\circ}\text{C})\,[\text{g}^{-1}]$ | | |
|---|---|---|---|---|---|---|
| | **Bark** | **Branch wood** | **Needles** | **Bark** | **Branch wood** | **Needles** |
| A | $1.4 \times 10^8$ | $2.8 \times 10^5$ | $6.5 \times 10^7$ | $3.8 \times 10^8$ | $1.6 \times 10^7$ | $1.4 \times 10^8$ |
| B | $1.1 \times 10^7$ | $6.7 \times 10^8$ | $3.1 \times 10^8$ | $7.5 \times 10^7$ | $1.8 \times 10^9$ | $1.4 \times 10^9$ |
| C | $1.1 \times 10^7$ | $2.7 \times 10^8$ | $1.8 \times 10^9$ | $9.4 \times 10^7$ | $9.2 \times 10^8$ | $7.3 \times 10^9$ |
| D | $9.8 \times 10^5$ | $1.3 \times 10^8$ | $1.3 \times 10^7$ | $1.1 \times 10^7$ | $1.0 \times 10^9$ | $1.3 \times 10^8$ |
| E | $4.9 \times 10^8$ | $6.6 \times 10^7$ | $1.9 \times 10^7$ | $1.3 \times 10^9$ | $3.0 \times 10^8$ | $2.8 \times 10^8$ |
| F | $2.4 \times 10^5$ | $5.9 \times 10^6$ | $9.8 \times 10^6$ | $3.2 \times 10^6$ | $6.9 \times 10^7$ | $1.8 \times 10^8$ |

**Table A2.** INM number concentration ($n_{INMs}$) on the surface of bark, branch wood and needles at −25 °C and −34 °C (see also Figure 4).

| Pine | $n_{INMs}(-25\,°\mathrm{C})\;[\mathrm{cm}^{-2}]$ | | | $n_{INMs}(-34\,°\mathrm{C})\;[\mathrm{cm}^{-2}]$ | | |
| | Bark | Branch wood | Needles | Bark | Branch wood | Needles |
|---|---|---|---|---|---|---|
| A | $2.6 \times 10^8$ | $2.8 \times 10^8$ | $5.6 \times 10^5$ | $1.7 \times 10^9$ | $5.2 \times 10^8$ | $3.6 \times 10^6$ |
| B | $4.8 \times 10^7$ | $1.7 \times 10^8$ | $5.8 \times 10^4$ | $2.0 \times 10^8$ | $4.5 \times 10^8$ | $7.8 \times 10^5$ |
| C | $2.1 \times 10^6$ | $1.4 \times 10^7$ | $1.4 \times 10^5$ | $9.9 \times 10^7$ | $6.5 \times 10^8$ | $4.1 \times 10^6$ |
| D | $2.7 \times 10^6$ | $6.4 \times 10^6$ | $6.3 \times 10^4$ | $5.2 \times 10^7$ | $2.2 \times 10^7$ | $1.3 \times 10^5$ |
| E | $6.7 \times 10^5$ | $1.6 \times 10^7$ | $2.3 \times 10^5$ | $1.4 \times 10^7$ | $5.0 \times 10^7$ | $4.6 \times 10^5$ |
| F | $6.1 \times 10^6$ | $3.0 \times 10^7$ | $4.9 \times 10^5$ | $5.6 \times 10^7$ | $6.2 \times 10^7$ | $1.1 \times 10^6$ |

**Table A3.** INM number concentration ($n_{INMs}$) of rain samples at −25 °C and −34 °C (see also Figure 5).

| Rain Sampler | $n_{INMs}(-25\,°\mathrm{C})\;[\mathrm{cm}^{-2}]$ | | | $n_{INMs}(-34\,°\mathrm{C})\;[\mathrm{cm}^{-2}]$ | | |
| | Pine A | Pine B | Pine C | Pine A | Pine B | Pine C |
|---|---|---|---|---|---|---|
| #1 | $2.6 \times 10^6$ | $5.5 \times 10^6$ | $1.7 \times 10^6$ | $1.8 \times 10^7$ | $3.7 \times 10^7$ | $7.3 \times 10^7$ |
| #2 | $2.4 \times 10^6$ | $5.3 \times 10^5$ | $1.7 \times 10^7$ | $2.4 \times 10^7$ | $2.1 \times 10^7$ | $9.6 \times 10^7$ |
| #3 | $1.6 \times 10^6$ | $2.4 \times 10^6$ | $1.3 \times 10^6$ | $4.1 \times 10^6$ | $5.7 \times 10^7$ | $3.0 \times 10^7$ |

**Appendix C. Estimation Calculation of INMs per m² Pine Stand**

To demonstrate the magnitude of the INM reservoir in a pine forest, we used our results to calculate a comprehensive estimate for the quantity of INMs contained within a single square meter of pine stand. In the first step, we calculated the bark surface area in a typical pine stand by assuming the stem as an open cylinder. Our results from the bark surface extracts (BSE), $n_{INMs}^{BSE}(-25\,°\mathrm{C})$ resemble an INM concentration per area bark. By multiplying the bark area of a typical pine stand with our result, we calculated the number of INMs, $n_{INMs}^{bark}$ per m² stand as follows:

$$n_{INMs}^{bark}(-25\,°\mathrm{C}) = n_{INMs}^{BSE}(-25\,°\mathrm{C}) \cdot (2r\pi h)\cdot d \tag{A1}$$

where $2r\pi h$ is the lateral area of an open cylinder, which is calculated using typical stem radii $r$ ($r_{min}$ = 11.8 cm, $r_{max}$ = 18.0 cm) and stem heights $h$ ($h_{min}$ = 8.7 m, $h_{max}$ = 15.9 m) for *Pinus sylvestris* [51,52]. The tree density $d$ ($d_{min}$ = 0.112 m$^{-2}$, $d_{max}$ = 0.2 m$^{-2}$) of pine stands was used to convert the INM number from a single tree to a m² pine stand, using the literature data [51,52].

Secondly, we used our results of the needle surface extracts (NSE), $n_{INMs}^{NSE}(-25\,°\mathrm{C})$ to estimate the number of INMs from needles, $n_{INMs}^{needles}(-25\,°\mathrm{C})$ per m² stand:

$$n_{INMs}^{needles} = 2 \cdot LAI \cdot n_{INMs}^{NSE}(-25\,°\mathrm{C}) \tag{A2}$$

where $LAI$ is the leaf area index and describes the one-sided leaf surface area per ground area for a particular tree or tree stand ($LAI_{min}$ = 1.48 m²$_{\text{needles}}$ per m$^{-2}$$_{\text{pine stand}}$, $LAI_{max}$ = 3.57 m²$_{\text{needles}}$ per m$^{-2}$$_{\text{pine stand}}$) [47–50]. We doubled the LAI to account for both sides since we assume the INMs can be found all over the needles' surface.

The sum of $n_{INMs}^{bark}(-25\,°\mathrm{C})$ and $n_{INMs}^{needles}(-25\,°\mathrm{C})$ is our estimation for the number of INMs per m² *Pinus sylvestris* stand, $n_{INMs}^{pine\ stand}(-25\,°\mathrm{C})$. We then used this number and an estimation for the area of *Pinus sylvestris* forest in Finland [53] to estimate the number of INMs from Finland $n_{INMs}^{Finland}(-25\,°\mathrm{C})$.

A few extra notes to this estimation:

- We did not account for branch wood, as the estimation for branch wood area of a tree in general is quite hard to find and uncertain. We also believe that this is to some

extent likely already accounted for in the LAI, since it is often an optical measure of how much light passes through a tree canopy and therefore branch wood likely plays a role in this. However, we decided to only use the needle surface concentrations in combination with the LAI.

- The surface roughness was also not accounted for. Still, we believe the likely underestimation by not accounting for roughness is partly compensated by the overestimation of the cylindrical shape we used for the bark area, as the trunk diameter likely decreases with height, and diameters are usually measured at one meter above the ground.
- Lastly, we decided to estimate a maximum to minimum range from our results and the literature data. We think this gives a better overview compared to a single average value.

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
