# Peer review of "Scots Pines (Pinus sylvestris) as Sources of Biological Ice-Nucleating Macromolecules (INMs)"

_atmosphere, doi:10.3390/atmos14020266_

Round 1

Reviewer 1 Report

A simple but adequate method to observe the ice nucleating property of biogenic particles present in pine trees is studied in this manuscript. The work is important in the context of increasing interest in the ability of ice nucleating bio-particles present in the atmosphere. After addressing the below-mentioned concerns and queries I can recommend the manuscript for publishing in the journal 'Atmosphere'. So I request a major revision here.

Major Comments:

1) Regarding Section 2 :

 a) In the method section, while describing the sample collection and storage, it is given that the collected samples were stored at -20 °C. So what was the temperature at which the samples were introduced into the instrument? This is applicable to all three sub-sections with powder, surface, and rain extracts.

b) Dilution of the powdered samples was discussed in the later stage even though no mention or details are given in the method section. Please add a sentence or two describing the dilution for all the samples (from A to F) and the final values used to reach Figure 3.

c)Rewrite the method section accordingly.

2) Figure 3:

We can see a similar trend and variation for samples D, E, and F (more pronounced for needles). And from the Appendix figures of the pine trees, we can infer that these are comparatively younger trees than the first 3. Will you be able to comment on the change in the INMs with the age of the pine trees? Or is there a reason for these similarities between the young pines?

3) A table in the Appendix section will be helpful, for the number concentration with T. As it was a bit difficult to deduce the values due to overlapping in the figures.

4)Rain Samples

The collection of rain samples was described with the direction from the tree, but it would be more helpful with the distance from the tree. Because as from Figure A6, it looks like they are at different distances from the tree. The use of directions can only be meaningful if you are discussing wind directions or patterns.

5) Figure 7 and 8: Are both the figures undiluted? Or is it a typo? And the explanation of Figure 8 is missing from the entire text.

6) Discussion:

a) Line 328: Is the 13° C referencing figure A.7? The two points for a single sample around 13 ° don't have any ice fraction. If the authors are referring to any other figures, then mention them here. 

b) Lines 370-382: Discussion missing details about the calculation elsewhere. What method is used in this calculation? other than the reference given. 

c) Re-write the discussuion part 

7) Conclusion:

Line 424-425: In lines 326-327 the ice nucleating temperatures were given as 17-28 and here it is 17-20, I guess these values represent the referred works. But the conclusion part is more for highlighting your results. So please rewrite the whole section accordingly.

Line 435: As a repetition of comment 6, since the values for a square meter of pine tree are reported here, more details on the calculation are important. Please include that in the method or discussion section.

Minor Suggestions:

1) Few minor spelling corrections are required throughout the manuscript,

for example, 'boreal' is misspelled as 'boral'. Line 95: IMM's instead of INM

Line 177: 'extracted surface area'?

Line 66: few cold hardy taxa? 

Line 227: 'if ' instead of 'of'

Line 254: A to concentrated ??

Line 404: 'arctic'   and 'source apportionment '

I have mentioned only a few here. A check for spelling and grammar correction is required.

.

Reviewer 2 Report

The topic is of interest to a wide group of readers. The language is appropriate, but the english can be improved particularly when it comes to technical terms that are not explained.

Round 2

Reviewer 1 Report

The authors have addressed the queries suggested and hence I recommend the manuscript be accepted in its present form.